# Analysis of Microbial Community in Circulating Cooling Water System of Coal Power Plant during Reagent Conversion

**Yichao Wang** [1,2], **Jiangyu Ye** [1,2,*], **Mingzhi Xu** [1,2], **Yunyi Li** [1,2] **and Jianjun Dou** [1,2]

1    College of Environment and Ecology, Chongqing University, Chongqing 400045, China;
a563186449@126.com (Y.W.); xmz_20126020@163.com (M.X.)
2    Key Laboratory of Three Gorges Reservoir Region's Eco-Environment, Ministry of Education,
Chongqing University, Chongqing 400045, China
\*    Correspondence: yejy@cqu.edu.cn; Tel.: +86-156-8341-8092

**Abstract:** The use of phosphorus-containing chemical corrosion and scale inhibitors has been found to result in excessive phosphorus discharge and an inability to reduce the high concentration of CODcr in the circulating cooling water, thereby making it challenging to comply with increasingly stringent sewage discharge standards. This study aims to assess the practicality of utilizing biological corrosion and scale inhibitors in coal power plants' operation, as well as investigating the correlation between water quality indicators and microbial communities during the conversion period. The data illustrates that, in comparison to the chemical method, there is a decrease in turbidity of the circulating water from 19.44 NTU to 9.60 NTU, a reduction in CODcr from 71.55 mg/L to 45.47 mg/L, and a drop in TP from 2.35 mg/L to 0.38 mg/L. Microbial community analysis during the transition period reveals that microorganisms rapidly establish a new equilibrium in the circulating water, sediment, and fiber ball, resulting in significantly different microbial community structures. The relative abundance of corrosive microorganisms such as *Flavobacterium*, *Pedomicrobium*, and *Hydrogenophaga* is significantly diminished in the circulating water, whilst the abundance of anaerobic microorganisms like *Anaerolineaceae* and *Rhodopseudomonas* in the sediment also declines. Conversely, there is an increased presence of microorganisms associated with contaminant degradation, such as CL500-3 and SM1A02. These findings suggest a decrease in the risk of system corrosion and an enhancement in contaminant degradation capability. This study provides evidence supporting the replacement of chemical agents with biological agents in circulating cooling water systems, contributing to more environmentally friendly and sustainable practices.

**Keywords:** coal power plant; circulating cooling water; corrosion and scale inhibition; microbial community structure

## 1. Introduction

In the industrial sector, circulating cooling water systems play a pivotal role, constituting 70–80% of the total water consumption and discharge [1]. The cooling process involves the spraying of water over packing material, which is then cooled through evaporation [2,3]. This process creates an optimal environment and nutrient balance conducive to the proliferation of microorganisms and the formation of biofilms [4]. Biofouling is a term used to describe the accumulation of deposits linked to the growth of living organisms such as bacteria, algae, and macro-organisms like sponges [5]. The production of extracellular polymer substances (EPS) by these organisms enhances the structural integrity of biofilms [6]. In scenarios where carbon sources and other electron donors are scarce, microbes can adapt by using elemental iron or other energetic metals as electron donors, leading to metal corrosion [7]. Elevated bacterial levels heighten the risk of microbiologically influenced corrosion (MIC) [8], while biofouling on wet surfaces can reduce the efficiency of heat exchangers and cause blockages in cooling circuit pipes [5], resulting in production loss and increased maintenance costs [4,9,10].

Chemical agents are traditionally employed to manage scaling, corrosion, and the proliferation of harmful microbes [11]. Agents such as polyphosphates, zinc salt, and organic phosphonic acid serve as scale and corrosion inhibitors, while sodium hypochlorite is typically utilized as a biocide [12]. However, these slimicides, including chlorine, phenylmercuric acetate, pentachlorophenol, tributyltin oxide, and isothiocyanates, pose a relative toxicity risk to humans. Furthermore, the introduction of biocides may inadvertently promote the corrosion of carbon steel [13]. In numerous regions globally, the law restricts high levels of inorganic phosphates [14]. The use of phosphorous scale inhibitors can exacerbate calcium phosphate deposition in circulating cooling water, presenting additional challenges [15]. Consequently, the increase in use of biocides has sparked a major scientific debate [16].

A multitude of research has delved into the potential of biological water treatment as a means to decrease CODcr and slime concentrations in circulating water systems [9,17]. Biofilms, by removing assimilable organic carbon (AOC) from process water, limit the primary substrate for biofouling microorganism growth [3]. Further studies have discovered that soluble extracellular polymeric substances (s-EPS) secreted by *Bacillus cereus* can form a biomineralized film, inhibiting corrosion and calcium scale crystal formation [18]. Concurrently, *Bacillus subtilis* has been found to form a dense biofilm on the surface of cold-rolled steel, decelerating corrosion [19]. Liu proposed the reduction of COD and nutrient limitation as strategies for controlling biological pollution without the need for increased bactericide usage [20]. In a different approach, bio-competitive exclusion (BE) strategies have been employed to curb the growth of sulfate-reducing bacteria (SRB) and MIC in the oil industry [21]. This study verified that there was a synergistic inhibition of SRB by adding nitrate-reducing bacteria (NRB), nitrate, nitrite, and molybdate to the system. Murshid suggested viewing biofouling in cooling water systems as a 'misplaced biofilm reactor' [22]. If the biofilm is correctly placed, nutrients could be absorbed in the water, safeguarding the system from corruption. These studies indicate that certain microorganisms or their secretions can influence scale and corrosion inhibition. In conclusion, biological methods may present a promising alternative with superior properties such as high efficiency, absence of secondary contamination, and cost-effectiveness compared with traditional chemical treatment. This underscores the potential of biological methods in revolutionizing the management of circulating cooling water systems.

While there is a wealth of research on the use of biological methods for controlling corrosion and fouling in circulating water systems, practical applications of biological water treatment remain limited. This study focuses on a thermal power plant in Tianjin with an installed capacity of $2 \times 500$ MW and a retained water volume of 5000 m$^3$. The plant has adopted the use of biological agents in its circulating water system, based on the ecological strategy of nutrient limitation. We conducted an in-depth analysis of the variations in water quality indicators and microbial communities. Furthermore, we evaluated the performance of the biological agents in inhibiting scaling, corrosion, and biofouling using a variety of methods. This study aims to bridge the gap between theoretical research and practical application, providing valuable insights into the potential of biological methods in the management of circulating cooling water systems.

## 2. Materials and Methods

### 2.1. Biological Agents and Carriers

The biological corrosion and scale inhibitors utilized were RJ-3 (refer to Table 1), supplied by Chongqing Rongji Environmental Company [23]. The total volume of the agent was 25 t, representing 0.05% of the water storage capacity of the circulating water system. The suspension fiber carriers were selected from commercial fiber carriers with a sphere diameter ranging from 25 to 30 mm. These carriers were procured from the Henan Ruibai Materials Company, China, and were fabricated from inert materials, serving solely as biocarriers. Further details of these carriers can be found in Table 2. Prior to the

experiments, all carriers underwent sterilization through autoclaving, ensuring they were free from any potential contaminants.

**Table 1.** Bacterial community in biological corrosion and scale inhibitors RJ-3.

| Bacterial Genus | Percentage |
| --- | --- |
| *Klebsiella* | 38.76% |
| *Prevotella* | 24.16% |
| *Streptococcus* | 7.89% |
| *Acinetobacter* | 5.59% |
| *Bacteroides* | 3.42% |
| *LactoBacillus* | 3.07% |
| *Enterococcus* | 3.43% |
| *PeptoStreptococcus* | 2.82% |
| *Kurthia* | 1.30% |
| Others | 10.56% |

**Table 2.** Information on biocarriers used in this study.

| Items | Fiber Carrier |
| --- | --- |
| Basic chemical composition | Polyethylene glycol terephthalate |
| Specification | 25–30 mm in sphere diameter |
| COD release capacity | Below the detection limit |
| Density | $1.38 \ kg/m^3$ |
| Specific surface area | $3000 \ m^2/g$ |
| Filling ratio | 0.3 |

*2.2. Sample Source*

To elucidate the dynamic shifts in microbial communities, samples of circulating water and fiber ball filler were procured from the circulating cooling water system on days 1, 4, 9, 14, and 19 post-agent addition. These samples were designated as W-1, W-4, W-9, W-14, W-19 and F-1, F-4, F-9, F-14, F-19, respectively. Sediment samples from the cooling water tank were collected during the chemical agent period and following the transition period, and were labeled as B-C and B-T, respectively. For each sample, three replicates were obtained to ensure data reliability. Prior to sampling, all equipment was subjected to high-temperature steam sterilization and subsequently air-dried for cooling. The sterilization process was conducted at 121 °C for a duration of 30 min to ensure complete decontamination. During the sampling process, sediment and fiber ball samples were securely packed and sealed within 100 mL sterile centrifuge tubes. Circulating water samples, ranging from 500–1000 mL, were collected using sterile polyethylene bottles. A 0.2 μm sterile filter membrane was employed to filter and sterilize the samples, with the filter membrane subsequently being packed into a 10 mL sterile centrifuge tube for sealing. All samples were systematically numbered and preserved in a refrigerator at −80 °C for subsequent analysis.

*2.3. System Transformation and Operation*

2.3.1. System Transformation

A stainless-steel formwork with fiber ball filler was installed in the collecting pool and fixed at the bottom of the pool. An appropriate amount of fiber ball fillers was added into the formwork. A sludge retaining wall with a height of 500 mm was also installed near the outlet in the collecting basin. Figure 1 shows a schematic diagram of system transformation.

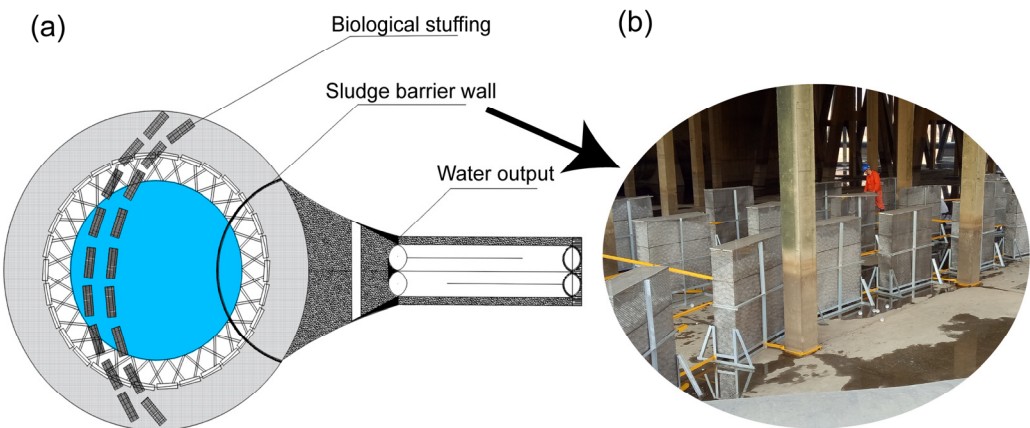

**Figure 1.** Diagram of system transformation. (**a**) Diagram of cooling tower structure. (**b**) Diagram of stainless-steel formwork with fiber ball filler.

### 2.3.2. System Operation

After the disinfection and sterilization system transformation of the circulating cooling water system technology, the cooling water system was sterilized. The compound bactericide was added to the collecting pool in a gradual and decentralized regional dosing-manner and continued to run for 24 h without circulating water blowdown. After sterilization, part of the cooling water was replaced with clean make-up water to reduce the residual chlorine concentration to below 0.01 mg/L. In the transitional stage, after the system completed disinfection and sterilization treatment, the biological corrosion and scale inhibitors could be added to start the experiment, and the use of all bactericides would be stopped during this period. To ensure safe and stable operation of the system at the initial stage of biological agent dosing, chemical corrosion and scale inhibitors were gradually reduced while adding biological agents over a transition period of 20 days.

### 2.4. Analysis Methods of Circulating Cooling Water

Turbidity, pH, $NH_4^+$-N, CODcr, TP, chloride ion, sulfate, total iron, hardness, and alkalinity of circulating cooling water sample were analyzed according to the methods in Table 3.

**Table 3.** Analysis methods of circulating cooling water [2].

| Water Quality Parameters | Experimental Methods and Standards |
|---|---|
| Turbidity (NTU) | Portable turbidity plan method |
| pH | pH indicator electrode |
| $NH_4^+$-N (mg/L) | Nessler's reagent spectrophotometry |
| CODcr (mg/L) | Dichromate method |
| TP (mg/L) | Ammonium molybdate spectrophotometry |
| $Cl^-$ (mg/L) | Nitrate radical titration method |
| $SO_4^{2-}$ (mg/L) | Gravimetric analysis |
| Total iron (mg/L) | Spectrophotometric o-phenanthroline |
| Calcium hardness (as mg/L $CaCO_3$) | EDTA titration |
| Total alkalinity (as mg/L $CaCO_3$) | Standard titration method |

### 2.5. PCR Amplification of Samples and Illumina MiSeq Platform Sequencing

PCR amplification was conducted using the primers 338F and 806R, which were specifically designed to amplify the variable regions V3 and V4 of the bacterial 16S rRNA gene [24]. The resulting PCR products were then subjected to analysis via 2.0% agarose gel electrophoresis, followed by size selection. Subsequently, the selected products were purified using an AxyPrep DNA gel extraction kit, a product of Axygen Biosciences, USA. Post-purification, the PCR products were quantified using QuantiFluor™-ST, a product of

Promega, USA. The quantified products were then pooled in equimolar concentrations and submitted to Majorbio Co., Ltd., located in Shanghai, China. Here, paired-end sequencing was performed on the pooled samples using the Illumina MiSeq platform.

*2.6. 16S Sequence Analysis of Samples*

Raw sequence reads from all samples underwent a series of processing steps. Initially, they were demultiplexed and quality filtered utilizing the Trimmomatic software 0.36, followed by merging using FLASH. Operational taxonomic units (OTUs) were then clustered based on a 97% similarity criterion using UPARSE 7.1, and chimeric sequences were removed using UCHIME. The high-quality sequences obtained were aligned against the SILVA 16S rRNA database. The classification of these sequences was performed using an RDP Bayesian classifier algorithm, with a threshold set at 70%. Principal coordinate analysis (PCA) was conducted with weighted UniFrac distances using Qiime 1.7.0 to assess the beta diversity among the samples. Rarefaction curves, and Shannon and chao diversity indices, were evaluated using Mothur 1.30.1 to estimate the alpha diversity within each sample.

Based on the relative abundance of OTUs, r2.4 software was used to generate Visual Circos analysis and Cladogram at the genus level to visualize the similarities and differences of dominant bacteria in sediment, circulating water, and fiber spheres. In addition, an analysis of similarities (Wilcoxon rank-sum test) was performed to test for significant differences between water microbial communities and sediments. To visually interpret community dissimilarity and investigate the relationship between microbial community data and environmental factors, a multivariate constrained ordination method was used. Redundancy analysis (RDA) was selected, and the significance of total physicochemical factors was tested with Monte Carlo permutations (permu = 999). Environmental factors were selected by the functions of envfit (permu = 999) and vif.cca, and the environmental factors with $p > 0.05$ or vif > 10 were removed from the following analysis. The vif values of pH, CODcr, $NH_4^+$-N, $Cu^{2+}$, TP, and Total iron were higher than 10 and removed. The analyses of heatmap and RDA were conducted in *R* for statistical computing using the vegan package.

## 3. Results and Discussion

*3.1. Comparison of Water Quality Stabilization Effects of Biological Agents and Chemical Agents*

As depicted in Figure 2a, after adding the biological corrosion and scale inhibitors, the $NH_4^+$-N in circulating water decreased significantly from 0.57 mg/L to 0.45 mg/L, reaching a minimum of 0.25 mg/L. In previous years, the use of chemical agents resulted in low microbial content in the circulating cooling water due to the application of bactericides, leading to an increase in $NH_4^+$-N levels in the circulating water with time. During the transition period, the addition of bactericides was halted. The introduction of biological corrosion and scale inhibitors fostered the growth of nitrifying bacteria in the circulating cooling water, facilitating the conversion of ammonia nitrogen into nitrate for removal. Contrary to previous years, Figure 2b,c demonstrate that there was no significant alteration in calcium hardness + total alkalinity and $SO_4^+ + Cl^-$, all of which comply with the requirements of the standard [25]. Figure 2d and Table 4 illustrate that the $Cl^-$ levels in the make-up water are lower than those in previous years, while the $Cl^-$ levels in the circulating water are higher than previous years, and the concentration ratio is also increased. Despite this, the system does not exhibit an increased tendency towards corrosion, suggesting that the biological agents exert a corrosion inhibition effect. Furthermore, the $Cl^-$ levels in the circulating water are regulated by the make-up water volume and the blowdown water volume during the actual operation process. The concentration multiple during the transition period increased by 0.86, leading to a reduction in the blowdown water volume and a consequent saving in water consumption. These findings underscore the water-saving effect of adding biological corrosion and scale inhibitors, highlighting their potential as a sustainable alternative to traditional chemical treatments.

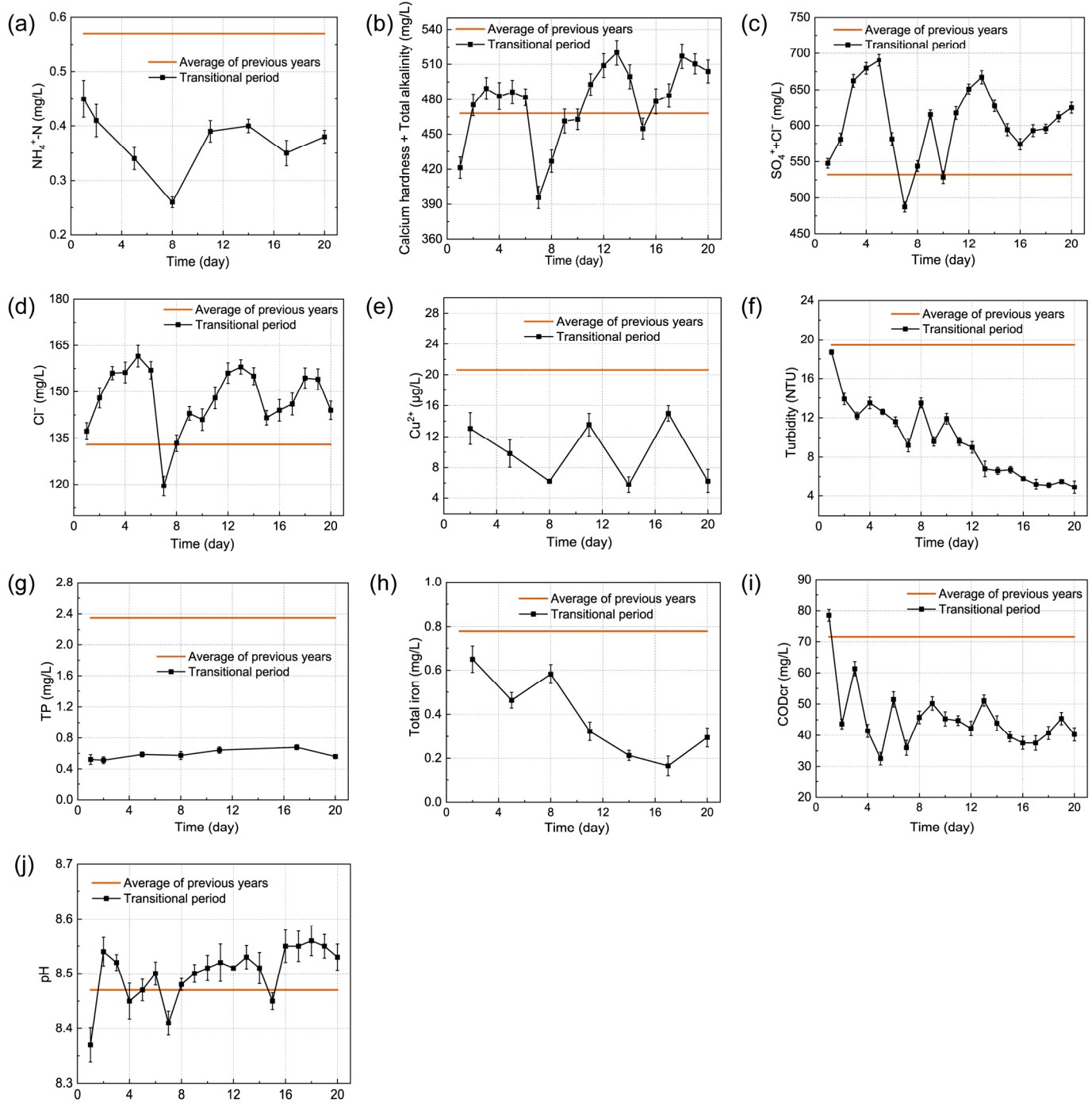

**Figure 2.** Comparison of water quality during biological agent dosing and chemical agent use in the same period in previous years. (**a**) $NH_4^+$-N, (**b**) calcium hardness and total alkalinity, (**c**) $SO_4^+$ and $Cl^-$, (**d**) $Cl^-$, (**e**) $Cu^{2+}$, (**f**) turbidity, (**g**) TP, (**h**) total iron, (**i**) CODcr, (**j**) pH.

Figure 2e–i illustrate that the biological agents demonstrate a superior effect in stabilizing water quality compared with the chemical agents used in previous years. A comparison with the chemical period reveals a significant reduction in turbidity from 19.44 NTU to 9.60 NTU, and a decrease in CODcr from 71.55 mg/L to 45.47 mg/L. This suggests that the biological corrosion and scale inhibitors are effective in eliminating pollutants in the circulating water. Simultaneously, the TP levels decreased from 2.35 mg/L to 0.38 mg/L, indicating a significant reduction in phosphorus emissions. Furthermore, the total iron and $Cu^{2+}$ concentrations in the system significantly decreased from 0.78 mg/L to 0.38 mg/L

and 20.64 µg/L to 9.95 µg/L, respectively, demonstrating the potent corrosion inhibition performance of the biological agents.

**Table 4.** Concentration of Cl⁻ and concentration times.

| Index | Interim Period | The Same Time in Previous Years | Variable Quantity |
|---|---|---|---|
| Average Cl⁻ in circulating water (mg/L) | 147.37 | 133.67 | 13.7 |
| Average Cl⁻ in make-up water (mg/L) | 38.35 | 43.6 | 5.25 |
| Concentration multiple | 3.97 | 3.11 | 0.86 |

*3.2. Analysis of Alpha Diversity and Beta Diversity of Microbial Community*

The rarefaction curves for all samples approached saturation, suggesting that the sequencing depth was sufficient for subsequent reliable analysis (Figure 3a). In the three types of samples, the dominant phylum of the microbial community varied slightly, with *Proteobacteria* emerging as the absolute dominant phylum. As depicted in Figure 3b,c, the Chao index and Shannon index of the suspended microbial community (W) during the transition period were higher than those during the chemical period. During the transition period, the Chao index of the suspended microbial community exhibited fluctuations and a decreasing trend, while the Shannon index initially increased before decreasing. This suggests a gradual decrease in the richness of the microbial community, coupled with an initial increase and subsequent decrease in diversity. The Chao and Shannon indices of the attached microbial community on the fiber ball filler (F) remained relatively stable. A comparison of the microbial communities in the sediment (B) during the chemical stage and at the end of the transition period, as shown in Figure 3d, revealed a decrease in both the Chao index and Shannon index. This indicates a decrease in the diversity of microbial communities in the sediment following the addition of biological corrosion and scale inhibitors. The dynamic trends of microbial community diversity within the system were further scrutinized. Figure 3e reveals that the bacterial communities in the 13 samples exhibited high diversity at the phylum level, with a total of 35 bacterial groups identified. These findings underscore the dynamic nature of microbial communities within circulating cooling water systems and the impact of transitioning from chemical to biological corrosion and scale inhibitors.

As illustrated in Figure 3f,g, the combined interpretation of PC1 and PC2 for the samples accounts for 78.42% of the total variance, effectively capturing the primary differences between the samples. The Principal Component Analysis (PCA) reveals that for both circulating water samples and fiber ball samples, there is a considerable distance between the microbial community structures at the initial and later stages of the transition period. However, the microbial communities in the circulating water and on the fiber ball filler exhibit relative stability during the later stages of the transition period [26]. The Non-Metric Multidimensional Scaling (NMDS) analysis corroborates this observation. It indicates that the microbial community structure is initially unstable upon the addition of biological corrosion and scale inhibitors. The circulating water system then undergoes a phase of establishing a new microbial ecological balance, followed by a gradual stabilization of the microbial community structure. Simultaneously, the NMDS analysis reveals that the microbial community structure in the sediment of the circulating water tank differs significantly from that in the circulating water and fiber filler. Moreover, both are markedly different from the structures observed during the chemical period. This suggests that the introduction of biological agents has substantially altered the microbial community within the system, underscoring the transformative potential of biological agents in managing circulating cooling water systems.

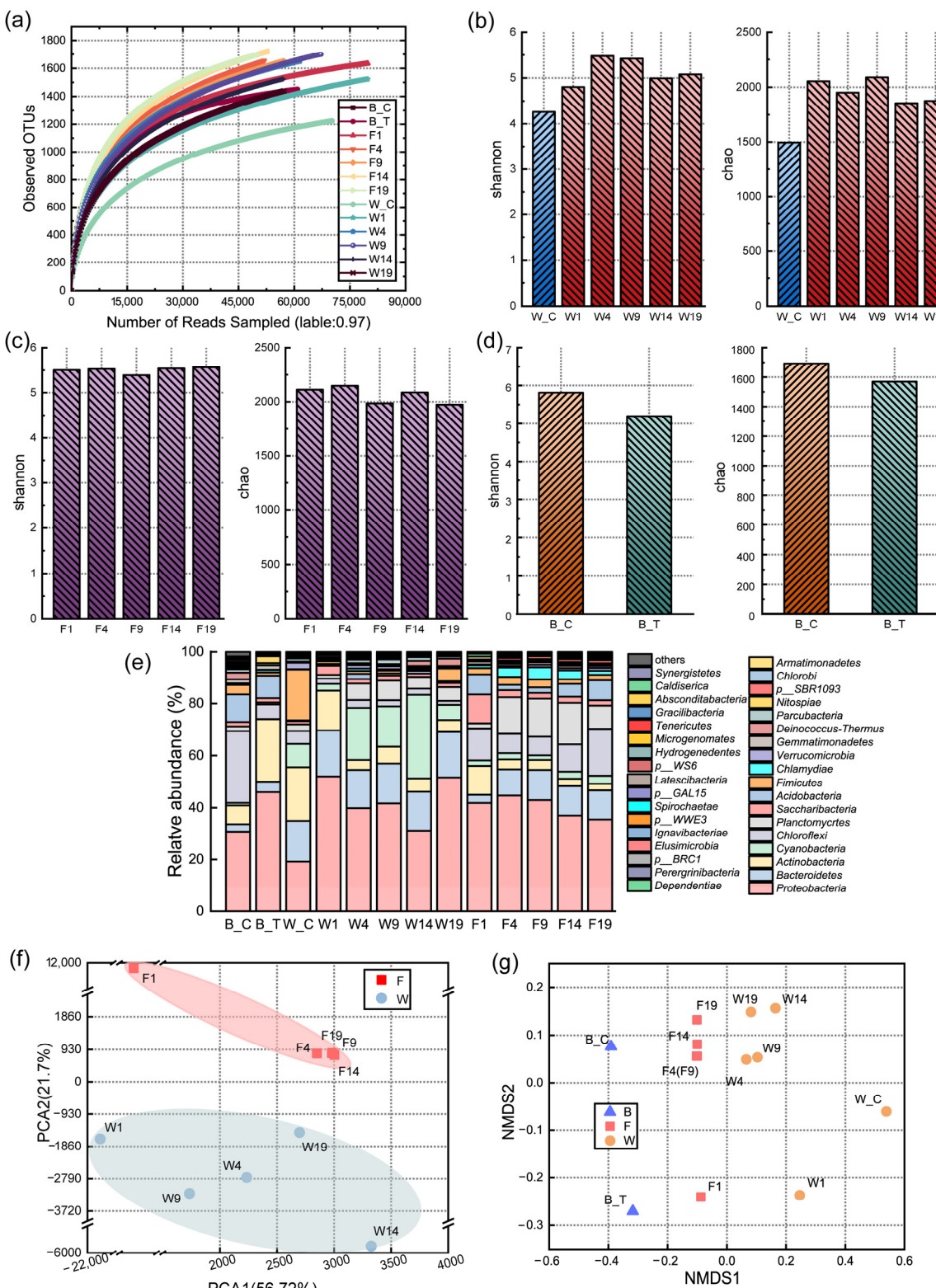

**Figure 3.** Analysis of alpha diversity and beta diversity of microbial community. (**a**) The observed OTUs of all the samples. (**b**–**d**) The Chao and Shannon indexes of different bacteria samples, B: sediment samples; W: free-living samples; F: fiber ball-attached samples. (**e**) Species composition at the phylum level. (**f**,**g**) Non-metric multi-dimensional scaling (NMDS) and principal component analysis (PCA) results of the microbial communities.

### 3.3. Similarities and Differences Analysis of Microbial Community Structure

The community structure of circulating water suspended microbes (W) at the genus level is depicted in Figure 4a. Dominant bacteria genera include *Hyphomicrobium, Polynucleobacter, Microcystis, Mycobacterium, hgcl_clade, Snowella, Flavobacterium, Mesorhizobium, Klebsiella, SM1A02, Hydrogenophaga, Rhodobacter, Deinococcus, CL500-3, CL500-29, Pedomicrobium, Sediminibacterium,* and *Methylobacterium.* Notably, *Comamonadaceae* (increasing from 1.36% to 8.09%) and *hgcl_clade* (increasing from 0.79% to 2.06%), which are dominant bacteria in W, are involved in nitrogen and phosphorus removal [27]. The increased abundance of these two bacteria enhances the removal of nitrogen and phosphorus. In addition, unique dominant bacteria genera in W like *Klebsiella, SM1A02,* and *CL500-29* also participate in nitrogen and phosphorus removal. *Pedomicrobium,* a ferric oxidizing bacterium, can adhere to various media surfaces and form biofilms [28]. *Sediminibacterium,* an iron-oxidizing bacteria, can oxidize $Fe^{2+}$ to $Fe^{3+}$ and can easily adsorb onto the surface of carbon steel, accelerating the pitting corrosion of carbon steel [29]. Following the addition of biological agents, the abundance of *Flavobacterium* (decreasing from 0.24% to 0.14%) and *Hyphomicrobium* (decreasing from 6.27% to 3.52%), which had exacerbated the corrosion of the system, significantly decreased. Over time, the abundance of bacteria *CL500-29* (increasing from 1.03% to 1.14%) and *Xanthomonadaceae* (increasing from 1.03% to 1.49%) associated with denitrification gradually increased. The abundance of the pathogen *Mycobacterium* was greatly reduced from 0.45% to 0.17%, reducing the pathogen infectivity. The relative abundance of *Microcystis* appreciated from 4.68% to 18.15% in the intermediate stage, indicating that the administration of the bacterial agent contributed to the degradation of pollutants and the improvement of water quality conditions, resulting in a significant increase in *Microcystis* [30].

The genus-level microbial community structure on the fiber ball packing (F) is depicted in Figure 4b. The dominant bacterial genera include *Hyphomicrobium, CL500-3, SM1A02, Neochlamydia, Mycobacterium, Rhodobacter, Stenotrophobacter, Reyranella, Hirschia, Mesorhizobium, Nakamurella,* and other unclassified genera. The relative abundance of these dominant genera was not high, with others accounting for more than 30%, indicating a complex structure of the attached microbial community. Interestingly, the dominant bacteria in the biological corrosion inhibitor did not dominate on the fiber ball. Among the dominant bacterial genera, *Hyphomicrobium* (decreasing from 8.89% to 4.58%), *Mycobacterium* (decreasing from 2.09% to 0.57%), *Rhodobacter* (decreasing from 1.43% to 0.82%), and *Mesorhizobium* (decreasing from 0.41% to 0.17%) were associated with nitrogen and phosphorus removal. *Rhodobacter* and *Mycobacterium* are also associated with the removal of organic matter. The abundance of these bacteria on the fibrous ball filler gradually decreased over time, indicating that the effect of biological agents was gradually weakened [31,32]. The relative abundance of *Neochlamydia* and opportunistic bacterium decreased gradually, which showed the inhibitory effect of biological agents on pathogenic microorganisms [33].

The community structure of bottom mud microbes (B) at the genus level is depicted in Figure 4c. B_C and B_T, respectively, represent the microbial community structure in the bottom mud of the cooling pool during the chemical agent period and at the end of the transition period. *Anaerolineaceae* and *Rhodopseudomonas,* which are strictly anaerobic microorganisms, exhibited a significant decrease in abundance from 15.95% to 1.75% and from 5.27% to 0, respectively, after the addition of biological corrosion and scale inhibition agents. This suggests a substantial reduction in the anaerobic environment in the sediment, thereby diminishing the anaerobic fermentation ability of microorganisms. Consequently, the abundance of the genus *Hydrogenophaga* decreased accordingly, from 0.96% to 0.72%. According to the research, *Micrococcusiaceae* has a significant negative correlation with the concentration of metal ions in water [34]. This explained that after the addition of biological corrosion and scale inhibitors, the concentration of metal ions in circulating water decreased and the corresponding abundance of *Microbacteriaceae* increased. It is worth mentioning that there was no *Hyphomicrobium* in RJ-3, but the addition of RJ-3 improved the overall

microbial structure of the system, eventually resulting in a significant increase in this genus in different samples, contributing to enhanced nitrogen removal from the water [35].

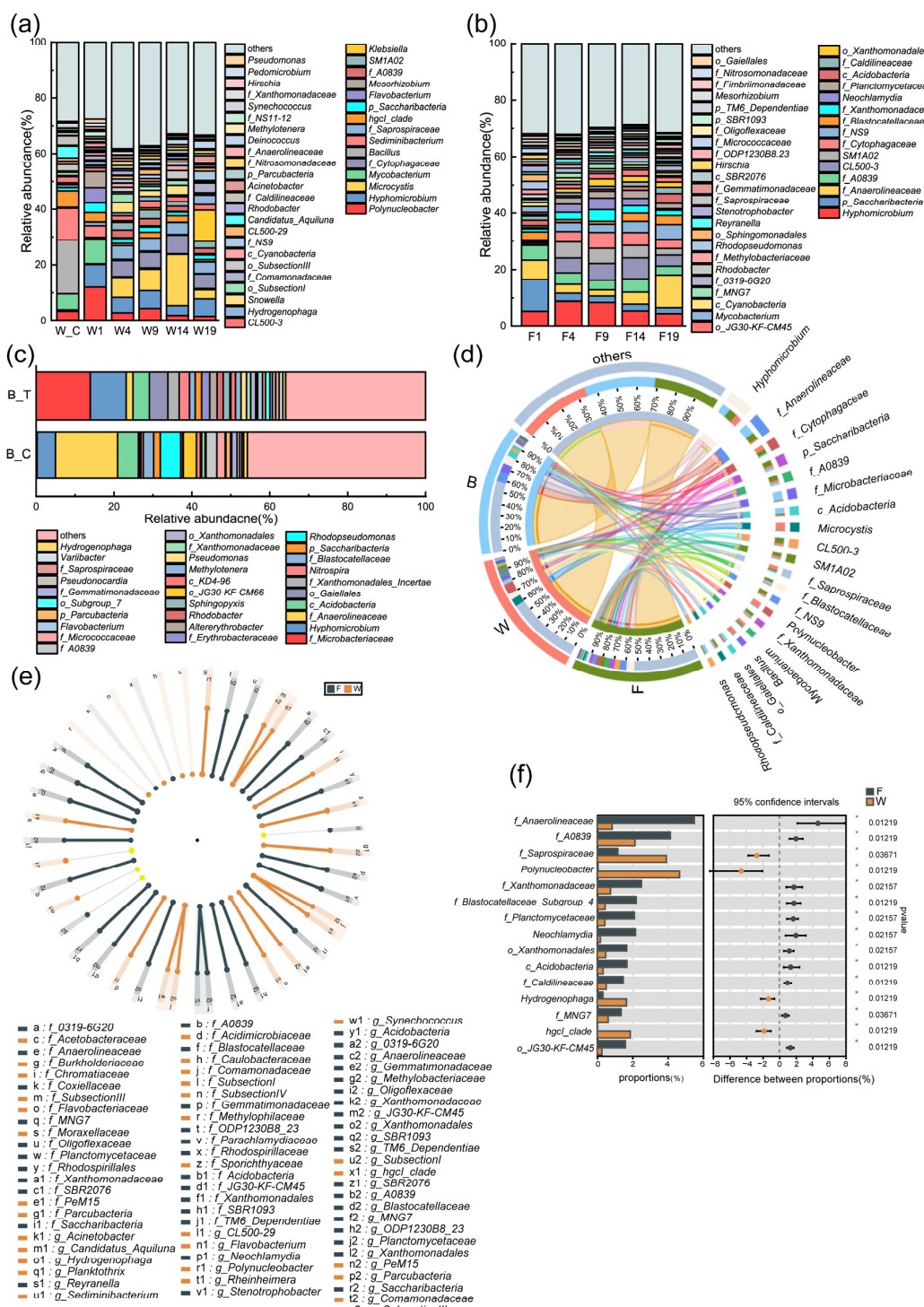

**Figure 4.** Similarities and differences in community composition and structure. (**a–c**) Bacterial community structure and distribution of free-living bacteria (W), fiber ball-attached bacteria (F), and sediment bacteria (B) at the genus level. (**d**) Visual Circos plot of the distribution of bacterial communities at the genus level for free-living samples (W), fiber-ball-attached samples (F), and sediment samples (B). The bar width of each genus represents its relative abundance in the corresponding sample. (**e**) Cladogram further shows the differences of microbial community between F and W samples. (**f**) Wilcoxon rank-sum test at the genus level. (* represents that the difference was significant (*p* < 0.05).)

Figure 4d–f illustrate the intergroup variances in microbial community composition of samples from free-living samples (W), fiber ball-attached samples (F), and sediment samples (B). Among the top 20 genera, *Hyphomicrobium* and *A0839* exhibit nearly identical relative abundances in recirculating water, fibrous pellets, and sediment, while the other species display considerable differences (Figure 4d). The abundance of *Saprospiraceae*, which has the ability to metabolize glucose, galactose, and specific proteins, is significantly higher in W than in F or B [36]. Conversely, the abundance of bacteria *Anaerolineaceae* in the fibrous bulb attachment samples was higher than that in the circulating water samples. Growth of *Anaerolineaceae* occurs under strictly anaerobic conditions, and all members of *Anaerolineaceae* are chemoheterotrophs, indicating the presence of an oxygen concentration gradient in the biofilm on the carrier, which is more conducive to pollutant removal [5]. *Acidobacteria*, which are enriched in the substrate and can utilize organosulfur compounds as a supplementary energy source [37], indicate that the corrosion capacity of the circulating water system can be reduced. This underscores biological agents' potential as a more effective and environmentally friendly alternative to traditional chemical treatments.

*3.4. Influence Analysis of Microbial Community Composition and Environmental Factors in Circulating Water System*

Spearman's correlation coefficient was employed to analyze the relationship between dominant functional bacteria in circulating water and water quality indices, as depicted in Figure 5a. The bacteria *Flavobacterium*, *Hyphomicrobium*, and *Hydrogenophaga* exhibited a proportional relationship with alkalinity. An increase in ammonia nitrogen provides more electron receptors for denitrification, promoting the growth of denitrifying bacteria, thereby adequately compensating for the alkalinity consumed by nitrifying bacteria. Bacteria related to alkalinity are primarily nitrifying and denitrifying bacteria. Nitrifying bacteria utilize alkalinity while denitrifying bacteria produce alkalinity. In anoxic environments, denitrifying bacteria utilize nitrate ($NO_3^-$) as a source of oxygen, reducing nitrate to nitrogen gas ($N_2$). This process generates bicarbonate, an alkaline agent, leading to an increase in alkalinity [38,39]. As seen from Figure 5a, *Comamonadaceae*, a denitrifying bacterium, displayed a positive correlation between its abundance in circulating water and alkalinity [40]. As shown in Figure 5b, the bacterial genera *Cyanobacteria*, which could generate organic acids and aggravates corrosion, are positively correlated with turbidity [41]. This suggests that an increase in turbidity in circulating water accelerates metal corrosion. Both in suspended and attached microorganisms, *Saccharibacteria*, an oral pathogenic bacterium [42], were positively correlated with turbidity. The addition of biological corrosion and scale inhibitors reduced turbidity and inhibited the proliferation of this pathogenic bacteria. Comparisons between Figure 5c,d reveal that the influence of environmental factors on suspended microorganisms is markedly greater than that on attached microorganisms. This indicates that the community structure of attached microorganisms is more stable. In addition, both graphs showed that the microbial community structure was greatly different from that in the middle and late stages of the agent dosing.

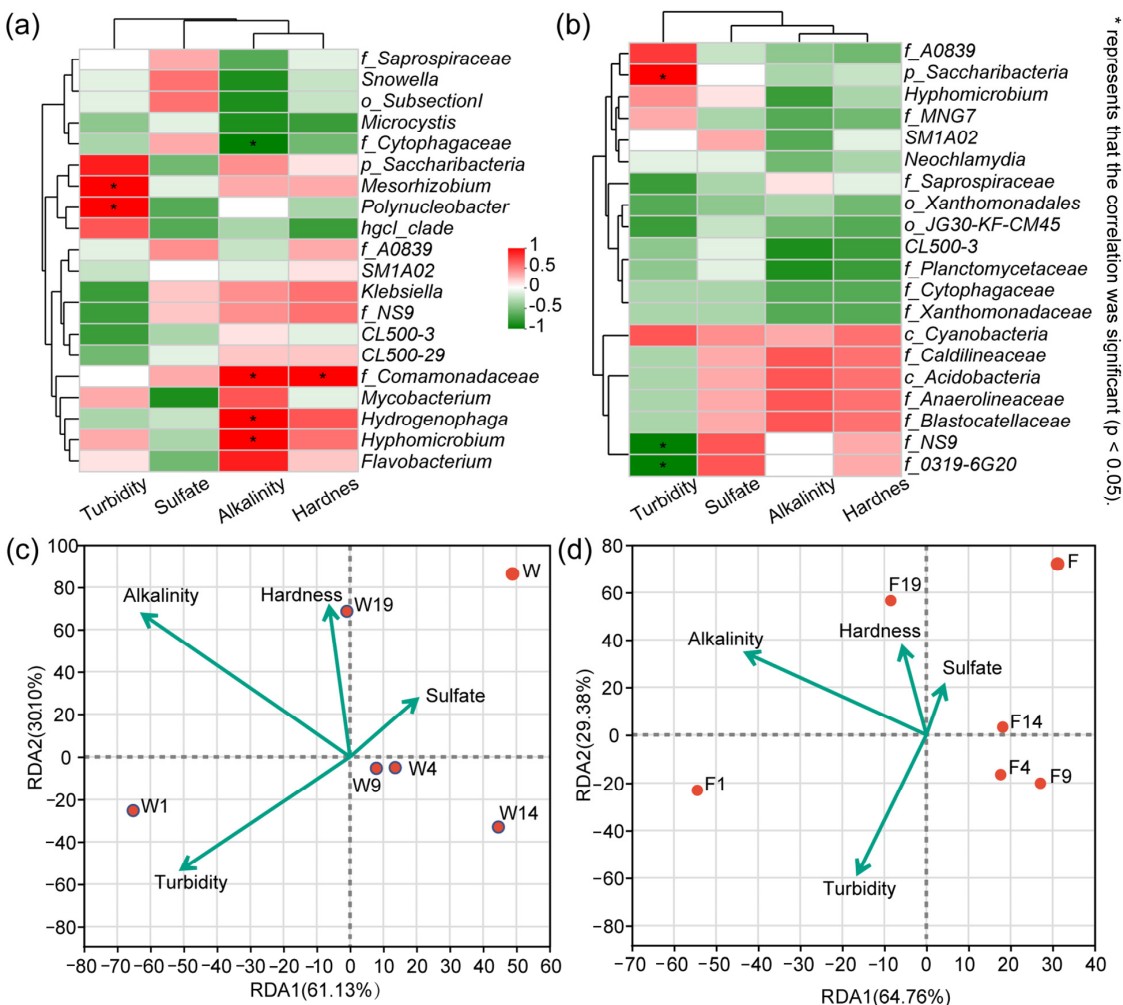

**Figure 5.** The interrelationship between environmental factors and microbial communities. (**a**,**b**) Heatmaps of the pairwise Spearman's rank correlations between environmental factors and samples. (**c**,**d**) Redundancy analysis (RDA) at the genus level. * represents a significant correlation.

## 4. Conclusions

Within a span of 20 days following the introduction of the biological corrosion and scale inhibitors into the circulating water system of a coal power plant, all water quality indices conformed to relevant standards. Notably, indices such as TP, COD, and total iron exhibited superior performance compared with the chemical method. Concurrently, the concentration multiple increased, facilitating water conservation. Microbial community analysis in the transitional system revealed significant differences in the microbial community structures of free-living bacteria, fiber ball-attached bacteria, and sediment bacteria. The dominant bacteria were primarily involved in nitrogen and phosphorus removal and organic degradation, and the relative abundance of corrosive microorganisms was significantly reduced. These findings demonstrate the ability of biological agents to stabilize water quality, inhibit corrosion and scale, conserve water, and reduce the secondary discharge of phosphorus pollution. This not only enhances the operational efficiency of coal power plants but also contributes to environmental sustainability. The use of biological agents in place of chemical ones marks a significant step towards more eco-friendly practices in the industry. It opens up new possibilities for managing water resources more effectively and reducing the environmental impact of power generation. This study underscores the transformative potential of biological agents and paves the way for further research into their application in various industrial settings. Additionally, studies could investigate

the long-term effects of biological agents on microbial community structures and system performance, providing deeper insights into their transformative potential.

**Author Contributions:** Y.W.: investigation, data curation, supervision, visualization, and writing—original draft. J.Y.: methodology, funding acquisition, conceptualization, and writing—review and editing. M.X.: visualization. Y.L.: methodology. J.D.: methodology and software. All authors have read and agreed to the published version of the manuscript.

**Funding:** This research was funded by the Ministry of Science and Technology of China, grant number 2013ZX07104-004-01.

**Institutional Review Board Statement:** Not applicable.

**Informed Consent Statement:** Not applicable.

**Data Availability Statement:** Data are contained within the article.

**Acknowledgments:** The financial support mentioned in the Funding part is gratefully acknowledged.

**Conflicts of Interest:** The authors declare no conflict of interest. The funders had no role in the design of the study; in the collection, analyses, or interpretation of data; in the writing of the manuscript; or in the decision to publish the results.

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
