# Peer review of "Analysis of Microbial Community in Circulating Cooling Water System of Coal Power Plant during Reagent Conversion"

_sustainability, doi:10.3390/su152316359_

Round 1
Reviewer 1 Report
Comments and Suggestions for Authors
The resolution of Figures 2, 3, 4, and 5 in this paper is so low that the reviewer cannot judge whether the experimental results support the conclusions of the manuscript. It is recommended that authors upload clear figures before submitting for review.
Author Response
请参阅附件。

Reviewer 2 Report
Comments and Suggestions for Authors
Your paper addresses an important issue related to the use of corrosion and scale inhibitors in coal power plants, specifically focusing on the feasibility of using biological agents as an alternative to chemical agents. Overall, your abstract and conclusion provide a clear overview of the study's objectives and findings. However, here are some suggestions for improvement:
Consider structuring your abstract to include separate sections for the background, methodology, results, and conclusions.
Figure2 is not clear.
Figure 3 is not clear too. All figures are good but not clear.
In your conclusion, emphasize the practical implications of your findings. Explain why the reduced concentration of corrosive microorganisms and improved water quality are significant for coal power plants and the environment.
Mention any potential areas for future research or further investigations related to this topic.
Author Response
请参阅附件。

Reviewer 3 Report
Comments and Suggestions for Authors
The manuscript entitled as “Analysis of Microbial Community in Circulating Cooling Water System of Coal Power Plant During Reagent Conversion”(Manuscript ID:sustainability-2655707), applied biological agents as promising approach for scale and corrosion inhibition and biofouling control in circulating water systems of thermal power plant in Tianjin, China, based on the ecological strategy of nutrient limitation, and made studies on the variations in water quality indicators and microbial communities during the conversion period to verify the feasibility of using biological corrosion and scale inhibitors in the practice of circulating cooling water treatment.
It is indicative from the results that compared with the chemical method, water quality indicators of circulating water after using RJ-3 biological agent have got improvement with the turbidity reduced from 19.44 NTU to 9.60 NTU, CODcr decreasing from 71.55 mg/L to 45.47 mg/L, and TP from 2.35 mg/L to 0.38 mg/L . Microbial community analysis results in the circulating water, sediment, and fiber ball during the transition period show that microorganisms quickly established a new balance on the circulating water and fiber ball, and the relative abundance of corrosive microorganisms such as Flavobacterium, Pedomicrobium and Hydrogenophaga is greatly reduced in the circulating water, the abundance of anaerobic microorganisms like Anaerolineaceae and Rhodopseudomonas in the sediment decreases significantly, while the abundance of microorganisms associated with contaminant degradation was increased, such as CL500-3、SM1A02, indicating a reduced risk for system corrosion with the increased contaminants degradation ability and the reliability to replace chemical agents with biological agents in circulating cooling water systems.
The manuscript needs to be further improvement with a minor revision before it can be accepted for publication. Comments and suggestion can be given as follows:
1. The quality for the figures needs to be further improved with the legends and font sizes to show more clearly (Fig 3, Fig4, Fig 5) and also the contrast between the curve and reference to be increased and leave more blank especially in fig2(f,g,h).
2. The text for discussing in lines 200-210 on page 7 should be in good accordance with the corresponding figures, readjust the order for the figures to keep them in good sequence with the text.
Xanthomonadaceae (from 1.03% to 1.49%) in lines 272 and Phycisphaeraceae (from 12.31% to 6.48%) in lines 285-286 can not be found in the exemplified dominant bacterial genera in the corresponding Figure 4(a) and Figure 4(b), respectively. The discuss text should keep in accordance with the figures to maintain the consistency.
Also, Pedomicrobium in line 333, page 11 can not be found in the figure 5(a,b) to maintain consistency between the discuss text and the figures.
3. Lines 175-176, page 5: “and the NH4+-N in the circulating water increased with concentration” is not very clear. Increased with whose concentration, added chemicals ? It should be better to change to: and the NH4+-N in the circulating water increased with time (or the use of chemical agent ) .
Lines 326-329, page 11: “and produces alkalinity” is deleted to keep the sentence logical and clear.
Lines 332-336, page 11: “metal corrosion and ”(line 335) is deleted to make the sentence more clear and logical.
Lines 330-331, page 11: “Nitrifying bacteria can use alkalinity while denitrifying bacteria produce alkalinity.” , why does denitrifying bacteria produce alkalinity? Give more explanation.
Lines 215-217,page 7: add “that” after “reveals” to follow the clause.
Lines 258-259, page 9: “as” changes to “are”.
4. Others typos and reference citation errors:
Note the reference citation format, complete the relevant information in references no.2, No.7, no 24, no25, no26, No38(2017,38(1):318-326.), No40 for the year, volume, issues , and page numbers from starting to ending. Label the chinese lterature citation in No 38 and No40 (in Chinese).
Repeated citations for references No.5 and No. 10; No. 4 and No.11; No 3 and No 19.
Comments on the Quality of English LanguageOverall, the English language is OK. It needs only a minor Editing.
Author Response
请参阅附件。

Reviewer 4 Report
Comments and Suggestions for Authors
This article is devoted to analysis of microbial community in circulating cooling water system of coal power plant. The authors studied the aqueous system with microorganisms for 20 days, which shows the large amount of work done. The results of this work confirm the feasibility of replacing chemical reagents with the use of microorganisms in cooling water circulation systems. This is possible because the risk of corrosion is reduced and the ability to degrade contaminants is increased. The use of microorganisms leads to improvements in many water quality indicators compared to the use of chemical methods. Furthermore, bacteria did not colonize in the recycled cooling water system.
The article is recommended for publication, but there are a few comments:
1) 9-10 “The use of phosphorus-containing chemical corrosion and scale inhibitors introduces a large amount of phosphorus, resulting in excessive phosphorus discharge.” (Maybe be replaced the second Phosphorus by Its)
2) 56,58 references should be in []
3) 77 should be m3
4) need to add captions to the fig 1 (what is a, what is b)
5) 173,175 182,183,185 Cl- check index
6) 182 extra dot between table and 4
7) Make references to figures in the text uniform (4(a), 5 (a))
8) improve the quality of figures

Author Response
请参阅附件。

Round 2
Reviewer 1 Report
Comments and Suggestions for Authors
The manuscript has been substantially improved besides the high resolution of figures, therefore I would recommend the publication of the manuscript after the following issues could be revised.
1. Line 99, there are repeated items, I suggest to delete "only were employed as inert carriers".
2. Table 3, please confirm the unit of alkalinity, it should not be described by the content of CaCO3.
3. Line 201, revise "alkalinitya".
4. Reference 10, please give the publication year. References 24 and 34, it should be "Chinese", instead of "Chinses".
Author Response
Reply to Reviewer #1
Dear Reviewers,
Thank you very much for taking the time to review this manuscript. These comments are very helpful to improve the quality of the manuscript. We have carefully revised out the manuscript. Please find the detailed responses below and the corrections highlighted in the re-submitted files.
Comments 1:
Line 99, there are repeated items, I suggest to delete "only were employed as inert carriers".
Response 1: Thank you for valuable comments. We have revised the sentence in lines 96-98: These carriers were procured from the Henan Ruibai Materials Company, China, and were fabricated from inert materials, serving solely as biocarriers.
Comments 2:
Table 3, please confirm the unit of alkalinity, it should not be described by the content of CaCO3.
Response 2: Thanks for pointing out our problem. We have reworked the units of alkalinity and hardness to (as mg/L CaCO3) in Table 3.
Alkalinity is a measure of the capacity of water to neutralize acids. This capacity is commonly attributed to bases such as hydroxides (OH-), carbonates (CO32-), and bicarbonates (HCO3-). All these species can react with hydrogen ions (H+) and thus, ‘neutralize acids’. Calcium carbonate (CaCO3) is often involved in these reactions, especially in natural waters. Therefore, for simplicity and consistency, alkalinity is reported as a concentration of CaCO3. In addition, the traditional, EPA-approved method to measure alkalinity is a titration at room temperature with a standard acid solution to a preselected end point. Results are expressed as mg/L CaCO3.
Comments 3:
Line 201, revise "alkalinitya".
Response 3: Thank you for your careful review pointing out the problems and we are very sorry for our careless mistake. We have changed “alkalinitya” to “alkalinity” on line 201.
Comments 4:
Reference 10, please give the publication year. References 24 and 34, it should be "Chinese", instead of "Chinses".
Response 4: Thank you again for the kind reminder. In our checking, we found that reference 10 and 5 were duplicated, and we have deleted reference 10. The Chinese citation labels in reference 24 and 34 have also been changed.